# Real-Time Measurement of Indoor PM Concentrations on Daily Change of Endocrine Disruptors in Urine Samples of New Mothers

**Dohyeong Kim [1] , Ju Hee Kim [2],* and SungChul Seo [3],***

[1]  School of Economic, Political and Policy Sciences, University of Texas at Dallas, Richardson, TX 75080, USA; dohyeong.kim@utdallas.edu
[2]  College of Nursing Science, Kyung Hee University, Seoul 02447, Korea
[3]  Department of Environmental Health and Safety, College of Health Industry, Eulji University, Seongnam 13135, Korea
*  Correspondence: juheekim@khu.ac.kr (J.H.K.); seo@eulji.ac.kr (S.S.)

**Abstract:** The recent innovation of IoT-based sensor technologies facilitates real-time monitoring of indoor air pollutants, such as particulate matter (PM), but its dynamic impacts on the level of endocrine disruptors in human body remain understudied. This feasibility study analyzed if the constant measurements of indoor PM concentrations collected at every five minutes are meaningfully associated with the levels of 15 types of endocrine disruptors in urine samples collected three times a day from nine new breastfeeding mothers in Seoul, Korea. Some promising results are observed in terms of detecting cumulative effects of $PM_{10}$ and $PM_{2.5}$ on some phthalate metabolites (MnBP, MiBP, MiNP, MCOP, MEOHP and MEHHP), BPA and TCS, at least for some participants. The findings from this study are expected to provide valuable directions for guiding future studies that discover potential associations between indoor PM concentrations and exposure to endocrine disruptors, which is still far from the consensus in the literature. Such efforts should offer empirical and scientific evidences for designing technology-based early warning/alarm services and evidence-based interventions to mitigate the level of exposure to PM and endocrine disruptors in their living environments.

**Keywords:** particulate matter; endocrine disruptors; urine samples; real-time measurement

---

Indoor air quality is one of the most critical indicators for economic and environmental sustainability [1]. Despite a growing evidence that indoor air pollutants are predominantly contributing to human exposure to a variety of chemicals due to the changes in lifestyle and indoor environments [2], a majority of studies still focus on the associations between outdoor particulate matter exposure and the level of endocrine disruptors in urine samples, such as phthalate metabolites, bisphenol A (BPA) and triclosan (TCS) [3,4]. Only a few researches have been dedicated to the effect of indoor air pollution on the exposure to those chemicals in commonly used exposure biomarkers, such as urine or blood, mostly based on a one-time collection of house dust samples [5,6]. This type of measurements could assess the long-term cumulative effects of indoor air pollutants, but fail to detect the dynamic temporal changes of those concentrations and their varying impacts on the level of endocrine disruptors in human body which should also fluctuate in a short interval.

The recent advancement of IoT (Internet of Things) based sensor technologies facilitates real-time constant monitoring of indoor air pollutants such as particulate matter (PM), volatile organic compounds (VOC) and radon [7]. Recently, such database has been constructed in a big data analytic framework and analyzed by a deep learning model in order to be utilized for practical prevention and intervention such as early warning or alert systems tailored to vulnerable households and individuals [8]. However,

an IoT-based air pollution monitoring and data analysis system have not been used to detect its association with the exposure to endocrine disruptors. This study aims to analyze the impacts of real-time measurements of indoor PM concentrations on exposure to various types of endocrine disruptors in urine samples from nine new breastfeeding mothers in Seoul, Korea who stay indoors most of the time, considering the growing concerns on health impacts of PM and chemical exposure in the country. This is a feasibility study with a relatively small sample size, but the findings and lessons from this study are expected to discover the dynamic and cumulative effects of PM concentrations on varying levels of exposure to different types of endocrine disruptors over time. The proposed research is expected to provide scientific and practical evidences for potential association between indoor PM concentrations and exposure to endocrine disruptors which is still far from the consensus in the literature.

The urine samples were collected in 10 to 28 June 2019 from ten new mothers living in Seoul, South Korea with their infant indoors for the majority of the time, but one sample was excluded due to metabolic disturbances or abnormal urine excretion (hematuria and urinary tract infection). Each of them was requested to take urine samples three times a day over one week (first morning voids, lunch time void, and bed time void) for which the collection time was recorded. The chemical analysis was conducted for all samples by analysis of ten phthalate metabolites (MnBP, MiBP, MEHP, MEOHP, MEHHP, MECPP, MEP, MiNP, MBzP and MCOP), BPA, TCS and three parabens (MP, EP and PP), following the previously reported methods for national biomonitoring programs [9,10]. Urine concentrations of environmental phenols were determined using a high-performance liquid chromatography–triple tandem mass detector (HPLC-MS/MS, API Triple Quad 550 System; AB SCIEX, Canada). The limit of detection (LOD) for each chemical is as follows: MnBP—0.282 μg/L; MiBP—0.188 μg/L; MEHP—0.139 μg/L; MEOHP—0.154 μg/L; MEHHP—0.139 μg/L; MECPP—0.113 μg/L; MEP—0.131 μg/L; MiNP—0.043 μg/L; MBzP—0.082 μg/L; MCOP—0.091 μg/L; BPA—0.02 μg/L; TCS—0.04 μg/L; MP—0.17 μg/L; EP—0.11 μg/L; PP—0.12 μg/L. For the values below LOD, we assigned the value of LOD divided by the square root of 2 [11].

Following a similar process of IoT-based indoor air monitoring for PM described in the previous study [8], a device with low-cost monitoring sensors were installed in each participant's dwelling to measure $PM_{10}$ and $PM_{2.5}$ concentrations as well as temperature and relative humidity every 10 min. The sampling device, PiCOHOME, is equipped with the PM sensor (PMSA003) and has been evaluated for first-class accuracy at an accredited certification facility in South Korea (Korea Testing & Research Institute) and calibrated before installation, as implemented in the previous work [8]. See Figure S1 for the calibration results. All data were stored real-time into the device and transferred to the cloud storage through Wi-Fi network. The data for indoor PM concentrations for each participant were then matched with those for endocrine disrupters based on time of measurement. The integrated data were used to assess bivariate association between three kinds of cumulative sum of PM concentrations (3 h, 6 h and 12 h) and each of 15 chemicals via time-series line plots and Pearson's correlation coefficients. This study was reviewed and approved by the Institutional Review Board of Kyung Hee University (KHSIRB-19-023).

Figure 1 shows the temporal trends of $PM_{2.5}$ concentrations and a specific chemical for some selected participants, for which the bivariate association is statistically significant. Despite some fluctuations and noise in the trend lines, some levels of correspondence are observable for MiBP, MiNP, BPA and TCS at least for some individuals during the one-week period. The matching trends show some promise for further investigation of possible association between indoor PM concentrations and exposure to at least some endocrine disruptors. See Figure S2 for the similar plot for $PM_{10}$. Table 1 shows for whom the cumulative effects of PM concentrations (3 h, 6 h and 12 h sum) on each of the endocrine disruptors are statistically significant. For all three cumulative levels, the $PM_{2.5}$ patterns are positively associated with MnBP, MiBP, MiNP, MCOP, BPA and TCS for four participants (C, D, G, H). The effects of $PM_{10}$ concentrations seem more noticeable for 12-h cumulative measures, which appear significant on MnBP, MiBP, MEOHP, MEHHP, MiNP, MCOP and TCS. The inconsistency of

significance across chemicals and participants may reflect different indoor environments and lifestyle, but also indicate how much chemical remains in urine after a certain number of hours of exposure.

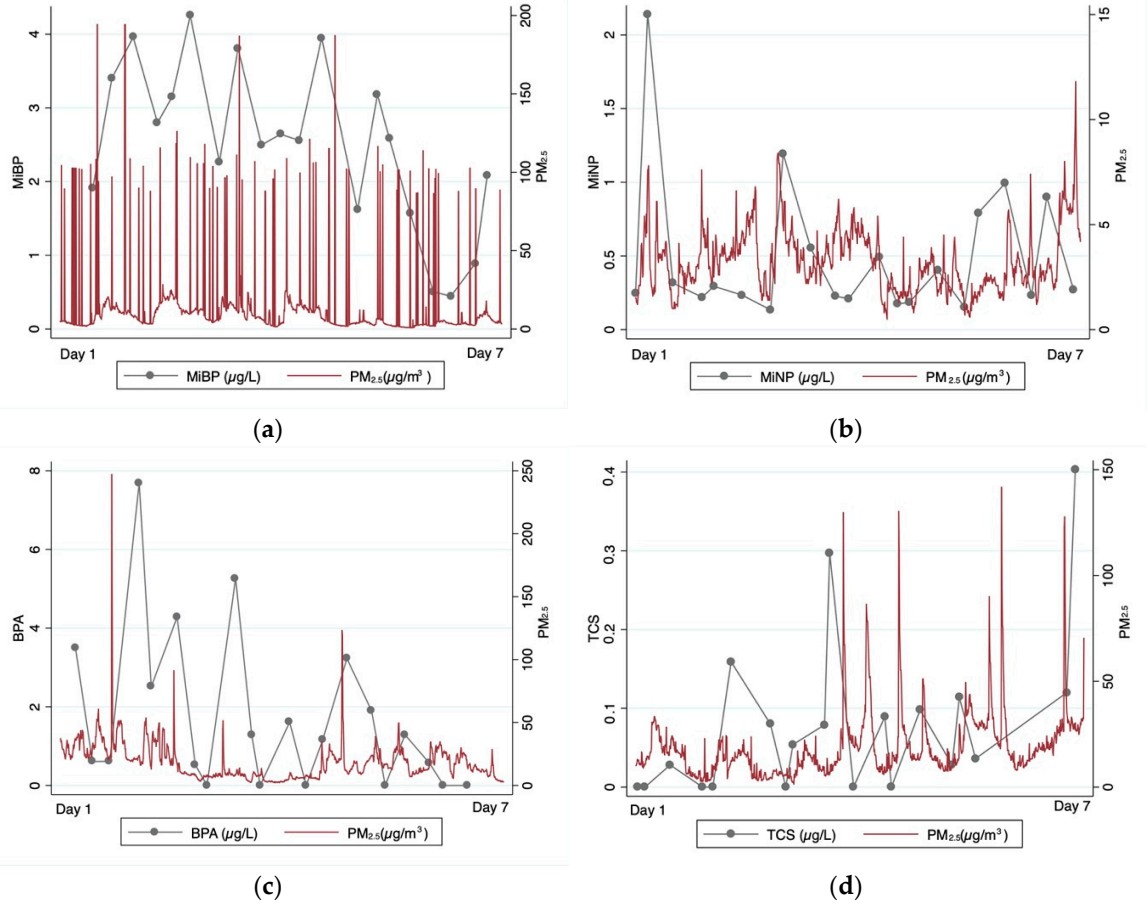

**Figure 1.** Time-series plots for $PM_{2.5}$ concentrations and a specific chemical for selected participants. (**a**) MiBP and $PM_{2.5}$ trends (for participant G). (**b**) MiNP and $PM_{2.5}$ trends (for participant D). (**c**) BPA and $PM_{2.5}$ trends (for participant H). (**d**) TCS and $PM_{2.5}$ trends (for participant C).

**Table 1.** Correlation between cumulative exposure to particulate matters ($PM_{2.5}$ and $PM_{10}$) and chemical concentrations in urine samples.

|  | $PM_{10}$ | | | $PM_{2.5}$ | | |
|---|---|---|---|---|---|---|
|  | 12-h Cumulative Sum | 6-h Cumulative Sum | 3-h Cumulative Sum | 12-h Cumulative Sum | 6-h Cumulative Sum | 3-h Cumulative Sum |
| MnBP | B, G |  | G | G | G | G |
| MiBP | B, G | C, G | C, G | G | G | G |
| MEHP |  |  |  |  |  |  |
| MEOHP | G |  |  |  |  |  |
| MEHHP | G |  |  |  |  |  |
| MECPP |  |  |  |  |  |  |
| MEP |  |  |  |  |  |  |
| MiNP | D, G |  |  | D | D | D |
| MBzP |  |  |  |  |  |  |
| MCOP | G |  |  | D | D | D |
| BPA |  |  |  | H | H | H |
| TCS | C | C |  | C | C | C |
| MP |  |  |  |  |  |  |
| EP |  |  |  |  |  |  |
| PP |  |  | E, F |  |  |  |

* The letters A to I indicate the unique ID for each participant, shown in this table only if the Pearson's correlation coefficient is positive and statistically significant ($p < 0.05$).

A significant bi-variate association between the dynamic and cumulative PM concentrations and the levels of endocrine disruptors is revealed at least for some phthalate metabolites, BPA or TCS, depending on the lifestyle and metabolism of new mothers. This is the first study that presents the role of the IoT-based real-time indoor PM monitoring tool on discovering its possible association with the levels of endocrine disruptors in urine samples. Despite several limitations, such as small sample size, we consider this investigation as a feasibility study that yields promising results and provides valuable directions for guiding future study. Future study should expand the database by incorporating more chemicals, outdoor PM concentrations and other covariates (other indoor environmental characteristics, lifestyle, diet, heathy behaviors, etc.), which can be analyzed via deep learning algorithms to predict the future patterns of endocrine disruptors based on spatiotemporal trends of both indoor and outdoor PM concentrations. Such efforts should provide empirical and scientific evidences for designing the models of technology-based early warning/alarm services or evidence-based educational programs tailored to vulnerable groups, such as new mothers and newborns, to mitigate the level of exposure to PM and endocrine disruptors in their living environments.

**Supplementary Materials:** The following are available online at http://www.mdpi.com/2071-1050/12/15/6166/s1, Figure S1: Calibration result (performance certificate and test result), Figure S2: Time-series plot for $PM_{10}$ concentrations and MiBP for participant G (similar pattern to Figure 1a).

**Author Contributions:** Conceptualization, D.K. and J.H.K.; methodology, D.K.; validation, D.K., J.H.K. and S.S.; formal analysis, D.K.; investigation, J.H.K.; resources, J.H.K. and S.S.; data curation, J.H.K. and S.S.; writing—original draft preparation, D.K., J.H.K. and S.S..; writing—review and editing, D.K.; visualization, D.K.; supervision, J.H.K. and S.S.; project administration, J.H.K.; funding acquisition, J.H.K. All authors have read and agreed to the published version of the manuscript.

**Funding:** This research was funded by the National Research Foundation of Korea (NRF) from the Korean Government (Ministry of Science, ICT) [grant number NRF-2018R1C1B6004256].

**Conflicts of Interest:** The authors declare no conflict of interest. The funders had no role in the design of the study; in the collection, analyses, or interpretation of data; in the writing of the manuscript, or in the decision to publish the results.

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
