# Peer review of "Real-Time Measurement of Indoor PM Concentrations on Daily Change of Endocrine Disruptors in Urine Samples of New Mothers"

_sustainability, doi:10.3390/su12156166_

Round 1

Reviewer 1 Report

The manuscript entitled “Real-time measurement of indoor PM concentrations on daily change of endocrine disruptors in urine samples of new mothers” by Kim et al. describes the correlation of particulate matters of indoor measurement with the urine samples. Few minor clarifications need to be explained.

  1. The authors provided the data for a few of the endocrine disruptors that are Phthalate metabolites, BPA, TCS, and parabens. Are there any other endocrine disruptors such as alkylphenols, synthetic musks, PCBs, cigarette residues in the rooms?  
  2. Before this study, the participants’ samples were collected and analyzed for the same endocrine disruptors (Baseline urine samples). If the participants had some previous exposure, and the sample collected may have that effect that needs to be ruled out.

Author Response

1. The authors provided the data for a few of the endocrine disruptors that are Phthalate metabolites, BPA, TCS, and parabens. Are there any other endocrine disruptors such as alkylphenols, synthetic musks, PCBs, cigarette residues in the rooms?  

The subjects of this study were infants and mothers. The chemicals we chose were the substances that these subjects are commonly exposed to within their home environments via household goods. We expect that the future study could include the materials mentioned by the reviewer for other groups or areas. Please see the highlighted lines 107-109 in the revised manuscript.

2. Before this study, the participants’ samples were collected and analyzed for the same endocrine disruptors (Baseline urine samples). If the participants had some previous exposure, and the sample collected may have that effect that needs to be ruled out.

We did not collect any baseline urine samples for the participants and so there is no way to measure the previous exposure. We could collect another sample for the same participants in the near future to detect the changes in exposure to the same endocrine disruptors by ruling out the baseline effects as you suggested.

Reviewer 2 Report

Dohyeong Kim et al studied levels of indoor PM concentrations and associated them with levels of 15 types of endocrine disruptors in urine  from 9 new breastfeeding mothers. The study is novel and the paper is well written and easily understandable. A weak point of the study is the low number of studied subjects and the lack of studied covariates. 

Some minor comments:

 Line 60: metabolic disturbances or abnormal urine excretion:Please specify.

Discussion: some other studies report harmful compounds like PBDEs and phtalates in dust, these might be worthwhile mentioning in the report. (science of the total environment 2018 Sep 1;635:817-827), (Chemospere.2019 Sep;231:216-224

Author Response

Dohyeong Kim et al studied levels of indoor PM concentrations and associated them with levels of 15 types of endocrine disruptors in urine from 9 new breastfeeding mothers. The study is novel and the paper is well written and easily understandable. A weak point of the study is the low number of studied subjects and the lack of studied covariates. 

We agree that the low number of study subject is the major limitation of the study, and actually plan to expand it by securing more funding after this feasibility. Actually, we collected many covariates for the study subjects but did not include the results to this paper as it focuses on showing a feasibility of linking the IoT-based real-time indoor PM measurements with a variety of endocrine disruptors (due to the word limit as well). We clarified such limitation and the direction for future study in the revised manuscript. Please see the highlighted lines 105-109 in the revised manuscript.

Some minor comments:

 Line 60: metabolic disturbances or abnormal urine excretion. Please specify.

We specified the line as “metabolic disturbances or abnormal urine excretion (hematuria and urinary tract infection).” Please see the highlighted line 59 in the revised manuscript.

Discussion: some other studies report harmful compounds like PBDEs and phtalates in dust, these might be worthwhile mentioning in the report. (science of the total environment 2018 Sep 1;635:817-827), (Chemospere.2019 Sep;231:216-224

Thanks for the suggestion. We have added the articles suggested by the reviewer to the Introduction section of the study. Please see the highlighted lines 34-37 in the revised manuscript.

Reviewer 3 Report

The authors present a communication associating real-time measurements of PM on endocrine disruptors in new mothers.

This preliminary study is well-written and described. My main concern is that the topic seems to be outside of the scope of Sustainability and may be better suited for a different journal. There are a few stylistic errors including not subscripting "2.5" in "PM2.5" or "10" in "PM10" and citations not being in a standard format, however the grammar and writing needs only light proofreading.

The color scheme chosen for Figure 1 is very bad. The lines (outside of the one with the circle) seem very similar and the authors are encouraged to use colors that are more different to better distinguish PM2.5 and PM10. The footnote describing the units is not necessary. The units should be placed directly on the axes labels. It may be more informative to show the same participant with different chemicals instead of a random assortment of participants in Figure 1.

While this is a good set of results for a pilot study, further development is needed to develop a full manuscript.

Author Response

The authors present a communication associating real-time measurements of PM on endocrine disruptors in new mothers. This preliminary study is well-written and described. My main concern is that the topic seems to be outside of the scope of Sustainability and may be better suited for a different journal.

Thanks for sharing your concern. We agree that our topic may be somewhat outside the journal scope, but believe that it perfectly fits with the Special Issue on “Air Pollution Monitoring and Environmental Sustainability”.

There are a few stylistic errors including not subscripting "2.5" in "PM2.5" or "10" in "PM10" and citations not being in a standard format, however the grammar and writing needs only light proofreading.

In response to the comments, we subscripted all PM2.5 and PM10 throughout the manuscript and corrected the citation format in the reference as required by the journal.

The color scheme chosen for Figure 1 is very bad. The lines (outside of the one with the circle) seem very similar and the authors are encouraged to use colors that are more different to better distinguish PM2.5 and PM10. The footnote describing the units is not necessary. The units should be placed directly on the axes labels. It may be more informative to show the same participant with different chemicals instead of a random assortment of participants in Figure 1.

Thanks for the suggestion. Following your suggestions, we have tried to change line colors in Figure 1 to make PM2.5 and PM10 more distinguishable to each other, but it did not much help due to a high similarity between them. Thus, we have decided to show only PM2.5 lines, which allows the figure to show temporal associations of PM and each chemical a lot clearer. We also placed the units directly on the axes as recommended. However, we have a specific reason for the selection of the participants (not a random assortment) since they are the pairs whose bivariate association is statistically significant. Please see the revised Figure 1 and the highlighted lines 80-81 in the revised manuscript.

While this is a good set of results for a pilot study, further development is needed to develop a full manuscript.

Thanks for your encouraging and constructive comments. We believe the revised manuscript has been substantially improved thanks to your comments and suggestions.

Round 2

Reviewer 3 Report

The authors have addressed most of the comments. The figures are still not subscripting "2.5" in "PM2.5" on the axes. Additionally, it may be useful for the authors to experiment with plotting PM10 as a dashed line in the figure to include it in the analysis if they wish to do so.

Author Response

Thanks for catching the oversight. We have revised Figure 1 to use a subscript for PM2.5 and superscript for m3. However, we have decided not to plot PM10 in the figure because it looks so busy and confusing even if a dashed line is used as you see in the figure as below. Hope you can understand our decision
